# Multi-Resolution Fusion: An Effective Approach to Anomaly Detection

## Abstract

Unsupervised anomaly detection has been profoundly impacted by the advent of large-scale Vision Foundation Models (VFMs). The prevailing paradigm leverages features from a pre-trained encoder, where anomalies manifest as statistical outliers. However, a fundamental challenge persists: industrial defects vary dramatically in scale, making it difficult for a single model to detect them all effectively. This paper introduces a critical insight: a "division of labor" exists between different input resolutions. We find that low-resolution inputs excel at robustly **recognizing** the presence of anomalies by capturing global context, while high-resolution inputs are essential for **refining** their segmentation boundaries with precision. To harness this synergy, we propose *Multi-Resolution Fusion* (*MRF*), a simple yet powerful training-free strategy. *MRF* constructs a feature pyramid from the input space by processing an image at multiple resolutions. By fusing features from this pyramid, our method, *MRF-AD*, effectively combines the recognition capabilities of low-resolution views with the refinement capabilities of high-resolution ones. Extensive experiments show that *MRF-AD* achieves highly competitive, and in several cases state-of-the-art, results on challenging benchmarks like MVTec AD 2, proving the efficacy of our multi-resolution approach.

## 1 Introduction

Unsupervised anomaly detection has been profoundly impacted by the advent of large-scale Vision Foundation Models (VFMs) like DINOv2 (Oquab et al., 2024). The paradigm is conceptually simple yet powerful: features from a pre-trained encoder are so descriptive of normal visual patterns that anomalies manifest as statistical outliers (Roth et al., 2022; Zhang et al., 2025). This training-free approach has become a cornerstone for industrial inspection, where annotating diverse defects is often impractical. However, a fundamental challenge persists: industrial anomalies vary dramatically in scale, from subtle, microscopic scratches to large, structural defects like missing components. Effectively detecting this wide spectrum of defects with a single model remains an open problem.

A common intuition is to use high-resolution inputs to capture fine-grained defects. However, contradict to common sense and previous finding in anomaly detection (Heckler-Kram et al., 2025), simply increasing the input resolution for ViT-based models does not guarantee better feature quality in anomaly detection setting. Recent studies (Fu et al., 2024) suggest that naively processing large images can lead to a degradation of patch feature quality, a phenomenon linked to the model's effective receptive field. We find that **low-resolution inputs excel at the core task of anomaly recognition**. By operating on larger patch sizes relative to the image, low-resolution views effectively capture global context, enabling them to robustly identify the presence of both large and small anomalies and reduce false positives in normal regions. However, their global perspective comes at the cost of localization precision, often producing coarse segmentation masks that misalign with the actual defect boundaries.

Conversely, **high-resolution inputs specialize in segmentation refinement**. As shown in Figure 1, they provide sharper, more accurate boundaries for detected anomalies. Yet, this focus on local detail comes at a cost: for large anomalies, the model can lose global context, resulting in "holes" within the segmented region where the anomaly is incorrectly classified as normal (Zheng et al., 2025). Therefore, the central challenge is not a simple trade-off, but rather how to effectively combine the

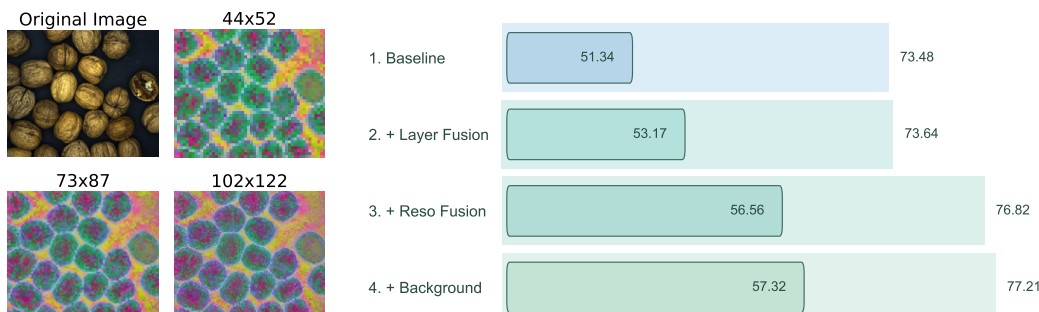

Figure 1: The "Recognition vs. Refinement" Dynamic. A PCA visualization of DINOv2 patch features. At lower resolutions (top left), the representation is globally coherent, enabling robust **recognition**. At higher resolutions (bottom right), boundary details are sharper, enabling precise **refinement**, but the object's interior becomes noisy, risking incomplete segmentation. Our work is motivated by synergizing these two roles.

Figure 2: Ablation of *MRF-AD*. We measure the performance of methods in different setting on $TEST_{pub}$ set of the MVTec AD 2 benchmark using AU-PRO$_{0.05}$ (inner bar) and AU-ROC$_{0.05}$ (outer bar). The baseline refers to using a single layer at maximum resolution to perform the anomaly detection. The layer fusion refers to using the combination of 7, 9, 11 layer of the same resolution. The resolution fusing refers to using 7, 9, 11 layer of embeddings extracted from 0.3-0.7 resolutions. The background refers to the introduction of BEN2-based background removal process. Our proposed Resolution Fusion greatly increased the performance in both AU-PRO$_{0.05}$ and AU-ROC$_{0.05}$.

robust *recognition* capabilities of low-resolution views with the precise *refinement* capabilities of high-resolution ones.

This observation leads to our central hypothesis: **an optimal anomaly segmentation requires a synergistic fusion of multi-resolution features.** Inspired by the success of pyramid-based representations in computer vision (Adelson et al., 1984; Lin et al., 2017), we introduce *Multi-Resolution Fusion* (*MRF*), a simple yet highly effective strategy designed for this purpose. Instead of building a feature pyramid from different network layers, our *MRF* strategy constructs it from the *input space* by processing the same image at multiple resolutions. It systematically and simultaneously leverages low-resolution features for robust **recognition** and high-resolution features for precise **refinement**. This creates a unified representation that is both sensitive in detection and precise in localization, shown in Figure 2.

Based on this strategy, we present *MRF-AD*, a training-free anomaly detection baseline that demonstrates state-of-the-art performance. Our main contributions are threefold:

1. We uncover a fundamental "division of labor" in VFM-based anomaly detection, where low-resolution inputs are primarily responsible for robust *recognition*, while high-resolution inputs are key for precise *refinement*.

2. We propose a simple and effective *Multi-Resolution Fusion* (*MRF*) strategy that explicitly harnesses this synergy, fusing features from multiple input resolutions to create a scale-robust representation without any training.

3. Our resulting method, *MRF-AD*, achieves highly competitive, and in several cases state-of-the-art, results on challenging benchmarks like MVTec AD 2 (Heckler-Kram et al., 2025), proving the efficacy of our approach.

## 2 RELATED WORK

Our work is situated at the intersection of unsupervised anomaly detection and the effective use of Vision Foundation Models (VFMs). We first review the primary paradigms in anomaly detection and

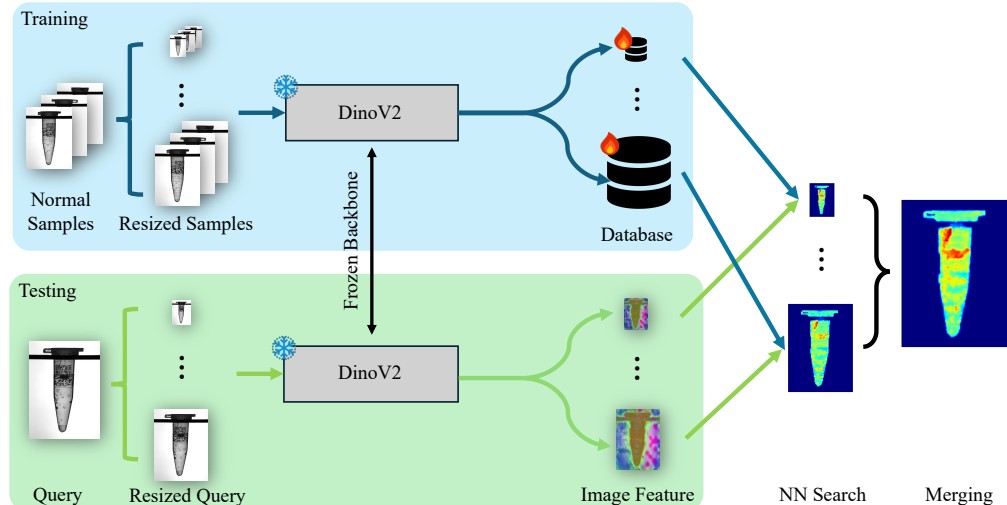

Figure 3: The overall pipeline of *MRF-AD*. The input image is resized into multiple resolutions and fed into a frozen vision foundation model (VFM) encoder. Features from different layers and **resolutions** are used to compute anomaly scores in parallel against dedicated memory banks. The resulting score maps are upsampled and fused via averaging to produce the final, scale-robust anomaly prediction.

then focus on the specific challenges and advancements related to leveraging VFM features for dense prediction tasks.

### 2.1 PARADIGMS IN UNSUPERVISED ANOMALY DETECTION

Unsupervised anomaly detection methods can be broadly categorized into three main families: reconstruction-based, synthesis-based, and embedding-based.

**Reconstruction and Synthesis-based Methods.** Reconstruction-based approaches operate on the principle that a model trained only on normal data will struggle to reconstruct anomalous inputs, thus yielding a high reconstruction error as an anomaly score. This is often realized through autoencoder or GAN architectures, with recent works extending to teacher-student frameworks like **RD** (Deng & Li, 2022) and **RD++** (Tien et al., 2023), or leveraging powerful generative models like in **DiffAD** (Zhang et al., 2023). Recently, **INP-Former** (Luo et al., 2025) utilizes DINOv2-Reg (Darcet et al., 2024) to encode the image and train an autoencoder on top of the feature embeddings to solve zero-shot and few-shot anomaly detection problems, and subsequent work such as **RoBiS** (Li et al., 2025) utilizes the same idea to solve full-shot problems. Synthesis-based methods, such as **CutPaste** (Li et al., 2021) and **SimpleNet** (Liu et al., 2023), take a different route by artificially creating defects to train a discriminative model to distinguish between normal and anomalous patterns. While effective, these methods often require extensive training or sophisticated data augmentation strategies, and show inferior generalization capability.

**Embedding-based Methods.** Our work belongs to the embedding-based family, which has become a dominant paradigm due to its training efficiency and high performance. These methods transform image patches into feature embeddings using a pre-trained network and measure anomaly by the distance of a test embedding to a memory bank of normal features. Early works like **PaDiM** (Defard et al., 2021) modeled normal feature distributions with Gaussians. The introduction of coreset-based memory banks in **PatchCore** (Roth et al., 2022) marked a significant leap, establishing nearest-neighbor search in the feature space as a powerful technique. The performance of these methods is fundamentally tied to the quality of the feature extractor. Consequently, the recent advent of VFMs like DINOv2 (Oquab et al., 2024) has spurred a new wave of research, with methods like **AnomalyDINO** (Damm et al., 2025) and **SuperAD** (Zhang et al., 2025) demonstrating state-of-the-art results by leveraging these powerful, pre-trained features. These VFM-based approaches form the direct context for our work.

## 2.2 Feature Representation and Scale in Vision Foundation Models

While VFMs provide exceptionally rich features, effectively translating them for high-resolution, pixel-perfect anomaly detection presents unique challenges, primarily concerning feature quality and scale adaptability.

**Improving Feature Quality.** A line of research focuses on improving the intrinsic quality of VFM features. It has been observed that standard ViT architectures may use certain patch tokens to store global information, polluting the local feature representations. To mitigate this, **DINOv2-Reg** (Darcet et al., 2024) introduced register tokens to serve as a dedicated "information sink", resulting in cleaner feature maps. Other works have also explored denoising and purification techniques to enhance feature fidelity (Jiang et al., 2025; Yang et al., 2024; Wang et al., 2024). These studies highlight that the raw output of VFMs is not always optimal and can be improved.

**Leveraging Multi-level and Multi-scale Features.** Another critical aspect is understanding and utilizing features from different network depths and spatial scales. It is well-established that shallow VFM layers capture fine-grained details, while deeper layers encode more abstract semantics (Lin et al., 2023; Jiang et al., 2023; Bolya et al., 2025). Methods like **SuperAD** (Zhang et al., 2025) exploit this by fusing features from multiple layers to create a more comprehensive representation. However, this only addresses feature diversity across network depth. The challenge of spatial scale, which we identify as the "resolution paradox", remains less explored. While some works aim to upscale VFM features to arbitrary resolutions (Fu et al., 2024; Suri et al., 2024; Couairon et al., 2025), they often require additional training, which harms their generalizability, and do not explicitly address the trade-off between high and low-resolution inputs for anomaly detection. Other works like $S^2$ Shi et al. (2025) use multi-resolution images as input, but it splits the large image into small sub-images, making it harder for model to process objects on the splitting boundaries. Our work fills this gap by proposing a training-free fusion strategy that explicitly addresses the "recognition vs. refinement" trade-off. By operating across the input resolution space, we synergize the strengths of different views to tackle the scale-variance problem in anomaly detection.

## 3 Preliminaries

### 3.1 Problem Definition

The primary goal of unsupervised anomaly segmentation is to identify regions within images that deviate from a learned distribution of normality. The model is trained exclusively on anomaly-free data.

Let $\mathcal{D}_{\text{train}} = \{\mathbf{x}_1, \mathbf{x}_2, \ldots, \mathbf{x}_N\}$ be the training set, where each $\mathbf{x}_i \in \mathbb{R}^{H \times W \times C}$ is a normal (anomaly-free) image. The test set, $\mathcal{D}_{\text{test}} = \{\mathbf{x}_1^*, \mathbf{x}_2^*, \ldots, \mathbf{x}_M^*\}$, consists of images that can be either normal or anomalous. For each test image $\mathbf{x}_j^*$, a pixel-level ground truth mask $\mathbf{M}_j^* \in \{0, 1\}^{H \times W}$ is provided, where a value of 1 indicates an anomalous pixel.

Given a test image $\mathbf{x}^*$, the task is to produce an **anomaly score map** $\hat{\mathbf{S}} \in \mathbb{R}^{H \times W}$. Each element $\hat{\mathbf{S}}_{uv}$ is an anomaly score for the pixel at location $(u, v)$, where a higher score signifies a greater likelihood of an anomaly.

### 3.2 The Embedding-based Anomaly Detection Paradigm

A dominant paradigm in this field is the embedding-based approach (Roth et al., 2022; Zhang et al., 2025), which typically operates in two main stages.

**1. Memory Bank Construction.** Given a generic feature extractor $\Phi(\cdot)$, a memory bank $\mathcal{M}$ is constructed from the normal training set $\mathcal{D}_{\text{train}}$. For each normal image $\mathbf{x}_i$, a feature map $\mathcal{F}_i = \Phi(\mathbf{x}_i)$ containing a set of patch-level feature vectors is extracted. The memory bank is the union of all such normal feature vectors:

$$\mathcal{M} = \bigcup_{\mathbf{x}_i \in \mathcal{D}_{\text{train}}} \{\mathbf{f} \mid \mathbf{f} \in \mathcal{F}_i\} \tag{1}$$

This memory bank $\mathcal{M}$ is sometimes reduced to a smaller coreset for efficiency.

**2. Inference and Scoring.** For a test image $\mathbf{x}^*$, its feature map $\mathcal{F}^* = \Phi(\mathbf{x}^*)$ is extracted. The anomaly score for each test feature vector $\mathbf{f}^* \in \mathcal{F}^*$ is computed by a scoring function, which measures the deviation of $\mathbf{f}^*$ from the distribution of normal features represented by the memory bank $\mathcal{M}$:

$$S(\mathbf{f}^*) = \text{Score}(\mathbf{f}^*, \mathcal{M}) \tag{2}$$

The specific implementation of the $\text{Score}(\cdot, \cdot)$ function varies across methods. For example, it can be the Mahalanobis distance to a Gaussian distribution fitted on $\mathcal{M}$ (Defard et al., 2021), or the $L_2$ distance to the nearest neighbor within $\mathcal{M}$ (Roth et al., 2022).

These patch-wise scores are then reassembled into a final anomaly score map $\hat{\mathbf{S}}$. Our work innovates upon this paradigm primarily by redesigning the feature extraction process $\Phi(\cdot)$ to be robust to scale variations, while adopting a well-established nearest-neighbor scoring function.

## 4 METHOD

### 4.1 OVERALL FRAMEWORK

To address the resolution paradox identified in the introduction, we propose *MRF-AD*, a simple yet effective training-free baseline for anomaly detection. Our approach redesigns the feature extraction stage of the classic embedding-based paradigm described in Section 3.2. As illustrated in Figure 3, our method consists of two core stages:

1. **Scale-Robust Feature Extraction:** We introduce a *Multi-Resolution Fusion* (*MRF*) strategy to build a comprehensive feature representation that captures information across multiple spatial scales and semantic levels.

2. **Anomaly Scoring and Fusion:** We compute anomaly scores against a set of dedicated memory banks and fuse the resulting score maps into a final prediction.

### 4.2 SCALE-ROBUST FEATURE EXTRACTION VIA MULTI-RESOLUTION FUSION

Our introduction established a fundamental limitation of VFM-based anomaly detection: the inherent trade-off between recognition and refinement tied to input resolution. To overcome this, our ***Multi-Resolution Fusion*** (***MRF***) approach constructs a comprehensive feature representation by building a resolution pyramid in the input space. This strategy allows us to explicitly combine the strengths of different views: low-resolution inputs provide the global context necessary for **robust anomaly recognition**, while high-resolution inputs offer the detailed information essential for **precise segmentation refinement**.

Specifically, for a given input image $\mathbf{x} \in \mathbb{R}^{H \times W \times C}$, we first generate a set of resized versions $\{\mathbf{x}_s\}_{s \in S_{\text{res}}}$, where $S_{\text{res}}$ is a predefined set of scaling factors. Each version $\mathbf{x}_s$ is then processed by a frozen DINOv2 encoder, which we denote as $\Phi$.

To further enrich the representation with hierarchical semantic information, we also extract features from multiple intermediate layers of the encoder, $S_{\text{layer}}$. This process yields a comprehensive collection of patch-level feature maps $\{\mathcal{F}_{l,s} \mid l \in S_{\text{layer}}, s \in S_{\text{res}}\}$, where each feature map is defined as:

$$\mathcal{F}_{l,s} = \Phi_l(\mathbf{x}_s) \in \mathbb{R}^{H_{l,s} \times W_{l,s} \times d} \tag{3}$$

Here, $\Phi_l$ represents the feature output at layer $l$, $(H_{l,s}, W_{l,s})$ are the spatial dimensions of the feature map for scale $s$, and $d$ is the feature dimension. Each feature map $\mathcal{F}_{l,s}$ thus captures a unique combination of spatial scale and semantic level.

### 4.3 ANOMALY SCORING AND MAP FUSION

Our scoring mechanism extends the nearest-neighbor approach to our multi-scale, multi-layer feature space, requiring a dedicated memory bank for each feature subspace.

**Training-free Setup: Memory Bank Construction.** We construct a dedicated memory bank $\mathcal{M}_{l,s}$ for each layer-resolution pair $(l, s)$. This is done by first extracting all feature maps $\{\mathcal{F}_{l,s}^{(i)}\}$ from the normal training images $\{\mathbf{x}_i \in \mathcal{D}_{\text{train}}\}$.

$$\mathcal{M}_{l,s} = \bigcup_{\mathbf{x}_i \in \mathcal{D}_{\text{train}}} \{\mathbf{f} \mid \mathbf{f} \in \mathcal{F}_{l,s}^{(i)}\} \tag{4}$$

**Inference Phase: Scoring and Fusion.** At inference time, for a given test image $\mathbf{x}^*$, we extract its feature maps $\{\mathcal{F}_{l,s}^*\}$ using the process from Section 4.2. For each feature map $\mathcal{F}_{l,s}^*$, we compute a corresponding anomaly score map $\hat{S}_{l,s}$ by scoring each of its feature vectors. The score for a feature vector $\mathbf{f}^* \in \mathcal{F}_{l,s}^*$ is its $L_2$ distance to the nearest neighbor in the dedicated memory bank $\mathcal{M}_{l,s}$:

$$S(\mathbf{f}^*) = \min_{\mathbf{f} \in \mathcal{M}_{l,s}} \|\mathbf{f}^* - \mathbf{f}\|_2 \tag{5}$$

Finally, to produce the single output anomaly score map $\hat{\mathbf{S}}$, we fuse all individual score maps $\{\hat{S}_{l,s}\}$. Each map is first up-sampled to the original image dimensions $(H, W)$ via bilinear interpolation, and then aggregated through element-wise averaging:

$$\hat{\mathbf{S}} = \frac{1}{|S_{\text{layer}}||S_{\text{res}}|} \sum_{l \in S_{\text{layer}}} \sum_{s \in S_{\text{res}}} \text{Upsample}(\hat{S}_{l,s}) \tag{6}$$

This fusion strategy ensures our final prediction benefits from both the global context of low-resolution views and the fine-grained details of high-resolution views, leading to a final prediction that leverages the **robust recognition** capabilities of low-resolution views and the **precise refinement** from high-resolution views, making it robust to anomalies of any scale.

## 5 EXPERIMENTS

In this section, we conduct a series of experiments to validate the effectiveness of our proposed method, *MRF-AD*. We first detail the experimental setup. Then, we present the main results comparing *MRF-AD* with state-of-the-art methods. Finally, we conduct comprehensive ablation studies to dissect our approach and empirically verify our central hypothesis regarding the "resolution paradox".

### 5.1 IMPLEMENTATION DETAILS

**Datasets.** We evaluate our method on the recent challenging anomaly detection benchmarks, MVTec AD 2 (Heckler-Kram et al., 2025), following the official evaluation protocol.

**Evaluation Metric.** Following the standard for recent benchmarks (Heckler-Kram et al., 2025), we use the pixel-level AU-PRO$_{0.05}$ as our primary evaluation metric. This metric appropriately handles anomalies of varying sizes and is not dominated by large anomalous regions. We also evaluate our approach using AU-ROC$_{0.05}$ as an additional metric.

**Preprocessing.** To ensure all images result in a reasonable number of tokens after resizing into different resolutions from 0.3 to 0.7, we resize images to have a comparable number of total pixels (approx. 5 million), while preserving their aspect ratios. The specific resolution conversions are detailed in Appendix A. Additionally, we employ a pre-trained segmentation model, BEN2 (Meyer & Spruyt, 2025), to remove background pixels before feature extraction, as this has been shown to improve performance. The impact of this step is analyzed in our ablation study (Section 5.3.3).

Additional implementation details can be found in Appendix D.

### 5.2 COMPARISON WITH STATE-OF-THE-ART METHODS

**Baselines.** We compare *MRF-AD* against several state-of-the-art methods, including PatchCore (Roth et al., 2022), RD4AD (Deng & Li, 2022), SimpleNet (Liu et al., 2023), and recent VFM-based methods SuperAD (Zhang et al., 2025) and RoBiS (Li et al., 2025). For fair comparison, we refer to the official results. For the evaluation of PatchCore, RD4AD and SimpleNet on *TEST$_{priv}$* and *TEST$_{priv,mix}$* dataset, we refer to their best results in the MVTec AD 2 benchmark report (Heckler-Kram et al., 2025). For the evaluation of SuperAD and RoBiS on those datasets, we refer to the results in the official leaderboard[1].

---

[1]MVTec AD 2 Official Leaderboard: https://benchmark.mvtec.com/

**Quantitative Results.** Table 1 presents the main results on the MVTec AD v2 $TEST_{priv}$ and $TEST_{priv,mix}$ datasets. Our method, *MRF-AD*, achieves highly competitive performance across the board. Notably, on the more challenging $TEST_{priv,mix}$ dataset, which includes out-of-distribution normal samples, *MRF-AD* achieves the best mean AU-PRO$_{0.05}$ score of **62.3%**, surpassing all other methods and demonstrating its superior robustness. While RoBiS shows a slightly higher mean score on the $TEST_{priv}$ set, our method achieves state-of-the-art results on several key categories like 'Can', 'Rice', and 'Sheet Metal'. We attribute this to our method's enhanced ability to handle anomalies of diverse scales, a direct benefit of our multi-resolution fusion strategy.

Table 1: Anomaly segmentation AU-PRO$_{0.05}$ performance (in %) on the MVTec AD v2 $TEST_{priv}$ / $TEST_{priv,mix}$ datasets. "Training" means if the method involves parameter tuning within a neural network. **Bold** indicates the best performance, and underline indicates the second best. On the challenging, out-of-distribution $TEST_{priv,mix}$ dataset, our method surpasses the second-best by a significant margin of 2.6%.

| Object | PatchCore | RD4AD | SimpleNet | SuperAD | RoBiS | *MRF-AD* (Ours) |
|---|---|---|---|---|---|---|
| Training? | ✗ | ✓ | ✓ | ✗ | ✓ | ✗ |
| Can | 12.8 / 10.2 | 15.0 / 13.7 | 21.9 / 3.2 | 33.7 / 27.8 | 30.3 / 20.0 | **39.0** / **30.1** |
| Fabric | 69.0 / 59.1 | **81.1** / 77.9 | 66.0 / 55.6 | 56.0 / 61.0 | 79.5 / **79.3** | 78.9 / 75.5 |
| Fruit Jelly | 71.5 / 70.8 | 54.9 / 54.7 | 64.9 / 63.0 | 72.2 / 71.6 | **74.5** / **74.1** | 66.9 / 66.5 |
| Rice | 47.8 / 28.9 | 27.4 / 26.2 | 21.9 / 12.3 | 54.2 / 55.7 | 62.3 / 63.9 | **65.4** / **65.8** |
| Sheet Metal | 72.4 / 54.0 | 54.2 / 51.4 | 46.8 / 34.1 | 70.5 / 68.4 | 75.5 / 73.5 | **78.2** / **74.8** |
| Vial | 75.8 / 72.2 | 69.9 / 67.7 | 56.1 / 38.8 | 75.7 / **73.0** | **76.8** / 69.6 | 74.8 / 70.3 |
| Wall Plugs | **68.4** / **53.7** | 55.1 / 39.5 | 25.6 / 10.5 | 50.8 / 41.2 | 62.2 / 24.8 | 44.3 / 36.2 |
| Walnuts | **80.4** / 71.6 | 71.7 / 64.6 | 62.7 / 55.7 | 76.4 / 75.9 | 77.1 / 72.0 | 80.2 / **79.6** |
| Mean | 62.3 / 52.6 | 53.7 / 49.5 | 45.7 / 34.1 | 61.2 / 59.3 | **67.3** / 59.7 | 66.0 / **62.3** |

We also evaluated our results using both AU-PRO$_{0.05}$ and AU-ROC$_{0.05}$ on $TEST_{pub}$ dataset, as presented in Table 2. More details are presented in Appendix B.

Table 2: Anomaly segmentation AU-PRO$_{0.05}$ (left) and AU-ROC$_{0.05}$(right) performance (in %) on the MVTec AD v2 $TEST_{pub}$ datasets. **Bold** indicates the best performance. Our method consistently outperforms baseline on different metrics.

| | AU-PRO$_{0.05}$ | | AU-ROC$_{0.05}$ | |
|---|---|---|---|---|
| | SuperAD | MRF-AD (Ours) | SuperAD | MRF-AD (Ours) |
| MVTec Ad 2 Mean | 54.7 | **57.3** | 76.7 | **77.2** |

**Qualitative Result.** As shown in Figure 4, the final "Merged" anomaly maps successfully combine the strengths of different input resolutions. Low-resolution views (e.g., x0.3) are adept at robustly identifying the presence of an anomaly, though the resulting detection masks are coarse. Conversely, high-resolution views (e.g., x0.7) provide sharp and precise boundaries for these anomalies. However, those views contain lots of noise or holes inside large anomaly areas (e.g., in walnuts). By fusing these different views, the MRF-AD method produces more accurate and complete segmentation masks than single-resolution approaches like SuperAD (Zhang et al., 2025).

## 5.3 ABLATION STUDIES AND ANALYSIS

We conduct a series of ablation studies on the MVTec AD v2 $TEST_{pub}$ subset to dissect our method and validate our core hypotheses.

### 5.3.1 ANALYSIS OF THE RESOLUTION PARADOX

Our primary motivation is the "resolution paradox": the observation that no single input resolution is optimal for all anomalies. To empirically validate this hypothesis, we evaluated the performance of our model using only a single input resolution at a time, across multiple feature layers. The results in

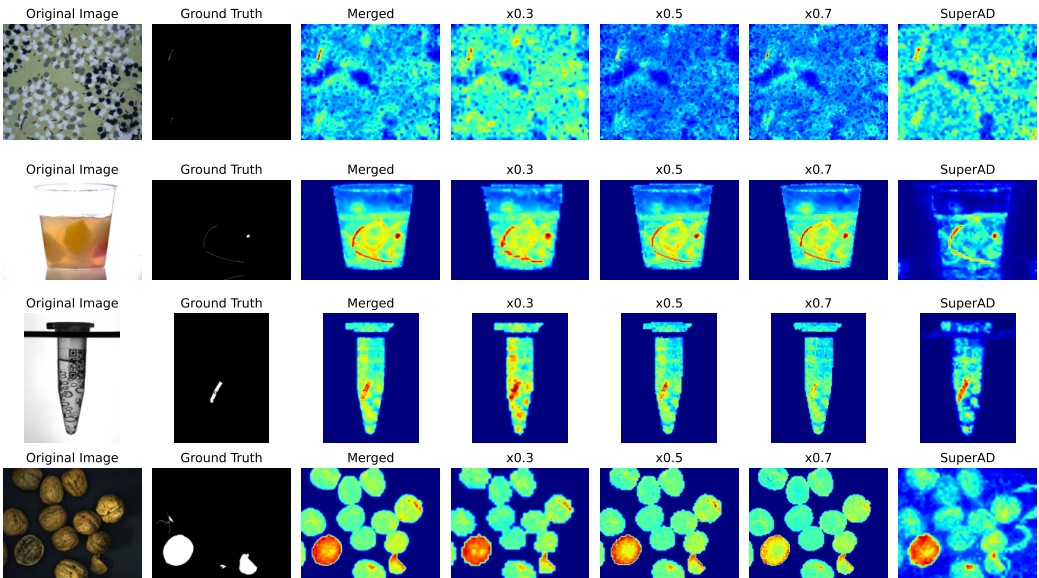

Figure 4: Qualitative comparison on the MVTec AD v2 dataset. Our merged result (Merged) successfully combines the robust detection from low-resolution views (e.g., ×0.3 correctly identifies the anomaly's presence but with a coarse mask) and the sharp boundaries from high-resolution views (e.g., ×0.7). This fusion leads to more accurate and complete segmentation masks compared to single-resolution methods and SuperAD.

Figure 5 (left) provide clear evidence for this paradox. For each layer (7, 9, and 11), the performance of the merged model (the rightmost bar in each group) consistently and significantly surpasses that of any single-resolution variant. This demonstrates that while individual resolutions may perform adequately on certain types of anomalies, they fail to achieve universal effectiveness. This finding underscores the fundamental limitation of single-scale approaches and establishes the necessity of our multi-resolution fusion strategy.

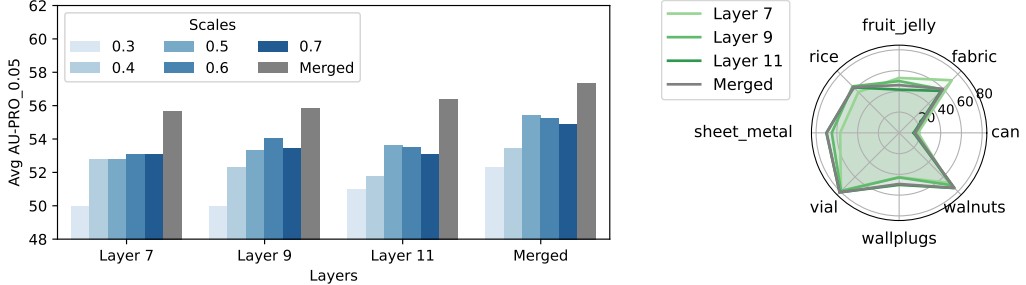

Figure 5: **Ablation studies on the necessity of multi-resolution and multi-layer fusion.** **(left)** Performance comparison of single-resolution models versus our multi-resolution fusion approach. Within each layer group (Layer 7, 9, 11), the "Merged" bar in gray (which fuses all resolutions) consistently outperforms any single-resolution bar (Scales 0.3-0.7). This validates our "resolution paradox" hypothesis and demonstrates the significant benefit of multi-resolution fusion. **(right)** Per-category performance breakdown for different feature layers, after multi-resolution fusion has been applied. No single layer (Layer 7, 9, or 11) is universally optimal across all categories. Our final model ("Merged", in red), which fuses all layers, achieves the most robust and superior performance, highlighting the complementary value of multi-layer fusion.

### 5.3.2 Effectiveness of Multi-Layer Fusion

Similarly, we study the impact of fusing features from multiple layers. It is well-understood that features from different network depths capture distinct information: shallow layers (e.g., layer 7) excel at representing fine-grained textures and local details, while deeper layers (e.g., layer 11) encode more abstract, semantic information (Lin et al., 2023; Jiang et al., 2023; Bolya et al., 2025). Our experiments in Figure 5 (right) confirm this trade-off; for instance, features from layer 7 perform well on 'fabric' (a texture-heavy category), while deeper features from layer 11 are better for 'walnuts'. This demonstrates that fusing features from different semantic levels is beneficial. This multi-layer fusion achieves the best average results, as demonstrated in Table 3, complements our core multi-resolution strategy, ensuring that at each input resolution, we leverage a comprehensive feature hierarchy from local details to global semantics.

Table 3: Ablation on layer fusion and background removal. Performance in AU-PRO$_{0.05}$ on $TEST_{pub}$ is evaluated. Fusing features from multiple layers yields better performance than using any single layer. Removing the background with BEN2 also provides a consistent, albeit marginal, improvement.

|  | 7+9+11 (Ours) | 7 only | 9 only | 11 only | w/o BG Removal |
|---|---|---|---|---|---|
| MVTec Ad 2 Mean | **57.32** | 55.66 ↓ −1.66 | 55.83 ↓ −1.49 | 56.39 ↓ −0.93 | 56.56 ↓ −0.76 |

### 5.3.3 Impact of Background Removal

Finally, we analyze the impact of our background removal preprocessing step. Table 3 shows that using BEN2 to remove the background provides a marginal but consistent improvement in the mean AU-PRO$_{0.05}$ score (from 56.56% to 57.32%). This suggests that focusing the feature extractor on the object of interest is beneficial, though not as critical as our core fusion strategy. We therefore include it in our final model configuration.

## 6 Conclusion

This paper addressed the critical challenge of detecting multi-scale anomalies in unsupervised anomaly detection. We identified and empirically validated a "resolution paradox", demonstrating that no single input resolution is optimal for all types of anomalies due to an inherent trade-off between recognition and refinement. To resolve this, we proposed *MRF-AD*, a simple and effective training-free method based on our *Multi-Resolution Fusion* (*MRF*) strategy. By systematically processing an image at various resolutions and fusing the resulting multi-layer features, our approach effectively combines the global context from low-resolution views for robust recognition with the fine-grained details from high-resolution views for precise segmentation. Experimental results on the MVTec AD v2 benchmark confirm our hypothesis, showing that our *MRF* strategy significantly outperforms single-scale approaches and achieves state-of-the-art performance, proving its robustness in detecting a wide spectrum of defects.

## 7 Limitations

Our proposed method, *MRF-AD*, demonstrates strong performance but has limitations. First, the *Multi-Resolution Fusion* (*MRF*) strategy's computational cost scales linearly with the number of resolutions processed. While effective, this may pose challenges for real-time applications requiring extremely low latency. Second, the performance of *MRF-AD* is inherently tied to the quality of the pretrained VFM. We observed a slight performance decrease when using DINOv2-Large, which we hypothesize is due to the nearest-neighbor search becoming less effective in its higher-dimensional (1024-D) feature space, or potentially due to increased noise in the embeddings. Finally, our reliance on a pretrained segmentation model (BEN2) for background removal introduces a potential failure point. If the segmentation model fails to generalize to new domains or incorrectly identifies the background, the performance of *MRF-AD* may be adversely affected.

## 8 REPRODUCIBILITY STATEMENT

Our method can be reproduced by following the experimental setup described in Section 5.1 and Appendix D. The pretrained DinoV2 model is publicly available via the HuggingFace platform. The MVTec AD 2 dataset can be obtained from MVTec Software upon acceptance of their usage agreement. The code used in our experiments will be released upon acceptance of the paper. Major results presented in this paper have been verified on the public leaderboard[2].

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

# A    INFORMATION ABOUT THE DATASET

## A.1    MVTEC AD V2

MVTec AD 2 (Heckler-Kram et al., 2025) is a recent challenging industrial anomaly detection benchmark that features great variation among normal samples, and various types of defects. One key aspect of the dataset is that it features high resolution and defects of various sizes, from extremely small to entire object, as shown in Figure 6.

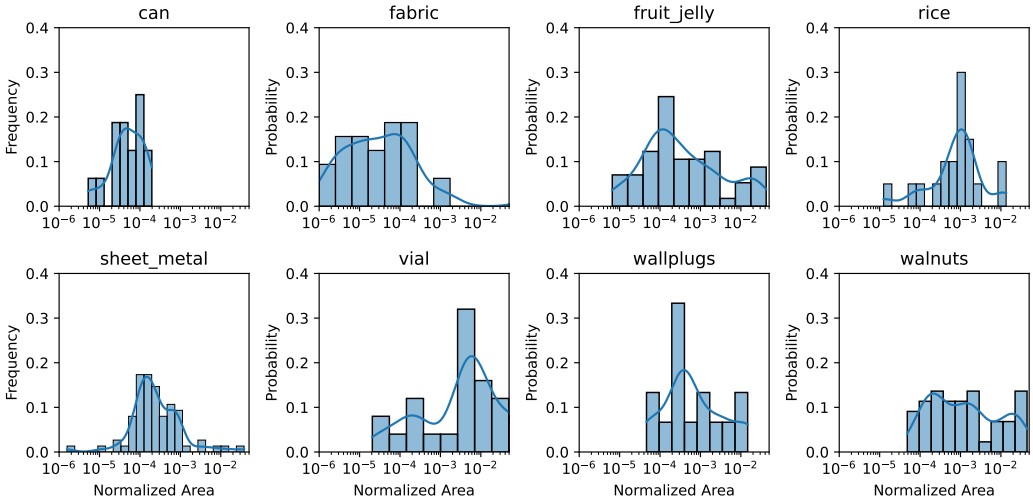

Figure 6: Distribution of defect areas for different categories in MVTec AD 2 Dataset. Majority anomaly areas are extremely small, only occupying less than 0.01% area, of to the original image size.

We also obtained the statistics of ratio of anomalies areas with respect to the original size, as presented in Table 4.

Table 4: Statistical analysis of the ratio (in $10^{-4}$) of anomalous regions to the original area in the test public subset of MVTec AD 2

| Object | Smallest | 1st quartile | 2nd quartile | 3rd quartile | Largest | Average | StDev |
|--------|----------|--------------|--------------|--------------|---------|---------|-------|
| Can | 0.05 | 0.29 | 0.53 | 1.04 | 1.97 | 0.70 | 0.55 |
| Fabric | 0.01 | 0.08 | 0.31 | 1.13 | 1837.00 | 58.67 | 319.41 |
| Fruit Jelly | 0.07 | 0.78 | 2.15 | 14.24 | 423.06 | 35.16 | 79.76 |
| Rice | 0.13 | 5.02 | 10.37 | 15.87 | 134.62 | 19.66 | 33.04 |
| Sheet Metal | 0.02 | 1.17 | 1.96 | 6.34 | 377.25 | 11.57 | 46.32 |
| Vial | 0.21 | 5.49 | 52.94 | 106.62 | 1296.88 | 169.92 | 312.11 |
| Wall Plugs | 0.46 | 2.55 | 3.85 | 30.95 | 147.32 | 26.91 | 46.74 |
| Walnuts | 0.48 | 2.13 | 11.03 | 68.24 | 443.49 | 62.38 | 103.45 |

To streamline the process in different resolution through a standard scaling factor, we preprocess images in different categories of the MVTec AD 2 dataset by resizing them to comparable total number of pixels, as shown in Table 5.

# B    ADDITIONAL EVALUATION

To fully evaluate *MRF-AD*, we also tested our method on the MVTec AD v2 $TEST_{pub}$ datasets using both AU-PRO$_{0.05}$and AU-ROC$_{0.05}$, as presented in Table 6 and Table 7, and compared it with the current reproducible SOTA SuperAD (Zhang et al., 2025). *MRF-AD*consistently performed better than existing method.

Table 5: Resolution conversion table for MVTec AD v2 categories to standardize the total number of pixels for our experiments.

| Object | Original Resolution | # of Pixels | New Resolution | New # of Pixels |
|---|---|---|---|---|
| Can | 1024x2232 | 2,285,568 | 1536x3348 | 5,142,528 |
| Fruit Jelly | 1520x2100 | 3,192,000 | 1900x2625 | 4,987,500 |
| Vial | 1900x1400 | 2,660,000 | 2470x1820 | 4,495,400 |
| *(... other categories with no change are omitted for brevity ...)* | | | | |

Table 6: Anomaly segmentation AU-PRO$_{0.05}$ performance (in %) on the MVTec AD v2 *TEST$_{pub}$* datasets. **Bold** indicates the best performance.

| Object | SuperAD (Zhang et al., 2025) | *MRF-AD* (Ours) |
|---|---|---|
| Can | 14.0 | **15.1** |
| Fabric | 48.8 | **59.9** |
| Fruit Jelly | **58.0** | 46.0 |
| Rice | 61.2 | **62.4** |
| Sheet Metal | 58.0 | **69.9** |
| Vial | 78.8 | **81.1** |
| Wall Plugs | 48.0 | **49.3** |
| Walnuts | 70.8 | **75.1** |
| Mean | 54.7 | **57.3** |

Table 7: Anomaly segmentation AU-ROC$_{0.05}$ performance (in %) on the MVTec AD v2 *TEST$_{pub}$* datasets. **Bold** indicates the best performance.

| Object | SuperAD (Zhang et al., 2025) | *MRF-AD* (Ours) |
|---|---|---|
| Can | 58.6 | **61.1** |
| Fabric | 68.3 | **71.7** |
| Fruit Jelly | **80.9** | 72.3 |
| Rice | 92.9 | **93.0** |
| Sheet Metal | 76.8 | **86.9** |
| Vial | 69.1 | **70.7** |
| Wall Plugs | **77.8** | 74.3 |
| Walnuts | **89.2** | 87.7 |
| Mean | 76.7 | **77.2** |

## C  ADDITIONAL ABLATION STUDY

We conducted an additional ablation study, presented alongside the main results in Table 8, which further supports the hypothesis that multi-resolution fusion leverages the strengths of both low and high resolutions. The results also confirm that no single layer is suitable for all types of anomalies.

## D  ADDITIONAL IMPLEMENTATION DETAILS

**Backbone and Hyperparameters.** We use the pre-trained DINOv2-Reg (Darcet et al., 2024) model with a ViT-B/14 architecture as our fixed feature extractor. For our *MRF-AD* method, the key hyperparameters are the set of input resolutions and the layers for feature extraction. Unless otherwise specified, we use a resolution set $S_{res} = \{0.3, 0.4, 0.5, 0.6, 0.7\}$ and a layer set $S_{layer} = \{7, 9, 11\}$. Features are extracted after the MLP block of each chosen layer.

**KNN Search.** We use the IndexIVFFlat approximate nearest neighbor search algorithm from Faiss (Johnson et al., 2019). For all experiments, we used $n_{list} = 512$ and $n_{probe} = 32$.

Table 8: *MRF-AD*'s performance in AU-PRO$_{0.05}$ on MVTec AD 2 *TEST$_{pub}$* dataset under multiple conditions. Background removal is applied consistently. $X - Y$ in the condition column means we are using features from layer(s) X and with input resolution(s) $Y$.

| Condition | can | fabric | fruit_jelly | rice | sheet_metal | vial | wallplugs | walnuts | avg |
|---|---|---|---|---|---|---|---|---|---|
| $7 - (0.3 + 0.4 + 0.5 + 0.6 + 0.7)$ | 18.14 | 71.66 | 52.95 | 5.58 | 56.84 | 79.28 | 42.67 | 68.22 | 55.66 |
| $9 - (0.3 + 0.4 + 0.5 + 0.6 + 0.7)$ | 15.78 | 59.21 | 49.85 | 63.31 | 64.89 | 79.21 | 42.96 | 71.41 | 55.82 |
| $11 - (0.3 + 0.4 + 0.5 + 0.6 + 0.7)$ | 13.94 | 57.17 | 41.60 | 61.91 | 70.08 | 81.10 | 50.14 | 75.19 | 56.39 |
| $(7 + 9 + 11) - 0.3$ | 12.55 | 55.23 | 42.24 | 59.31 | 62.87 | 80.23 | 37.72 | 68.20 | 52.29 |
| $(7 + 9 + 11) - 0.4$ | 14.35 | 47.95 | 42.89 | 61.31 | 61.07 | 78.89 | 48.45 | 72.47 | 53.42 |
| $(7 + 9 + 11) - 0.5$ | 16.23 | 55.14 | 46.03 | 57.66 | 62.87 | 80.08 | 49.52 | 75.57 | 55.38 |
| $(7 + 9 + 11) - 0.6$ | 17.78 | 59.22 | 46.60 | 56.60 | 60.61 | 78.77 | 46.98 | 75.29 | 55.23 |
| $(7 + 9 + 11) - 0.7$ | 14.54 | 58.77 | 47.76 | 54.95 | 62.18 | 76.97 | 49.95 | 73.81 | 54.86 |

**Hardware.** All experiments were conducted on a single server equipped with an NVIDIA A100-SXM4-80GB GPU and two AMD EPYC 7713 64-Core Processors. A full run of our method on a single MVTec AD v2 category takes less than 3 hours.

# E    THE USE OF LLMS IN THIS RESEARCH

LLMs are used to find related work and to help write and polish this paper.

