# OpenReview forum: "Multi-Resolution Fusion: An Effective Approach to Anomaly Detection"
_ICLR.cc/2026/Conference — ICLR 2026 Conference Withdrawn Submission_

### Official Review · Reviewer_KLGT · 2025-10-27

**Soundness:** 2
**Presentation:** 2
**Contribution:** 1
**Rating:** 2
**Confidence:** 3

**Summary:**

This paper introduces a training-free anomaly detection method that leverages multi-scale DINOv2 features as a database. During inference, the method performs nearest-neighbor search to localize anomalies in industrial images. The reported results show comparable performance to prior works.

**Strengths:**

1. The approach is conceptually simple and easy to follow.

2. The paper provides initial analyses on the visual and statistical effects of multi-scale inputs, which effectively convey the motivation.

**Weaknesses:**

1. The work lacks sufficient novelty and insight. Building databases from multi-layer encoder features has been extensively explored in previous studies since 2020. Feeding images at multiple resolutions alone does not constitute a substantial technical advance for ICLR.

2. Using multiple resolutions increases memory requirements and inference cost due to heavier nearest-neighbor search operations.

3. The results in Table 1 show only marginal improvements and do not demonstrate a clear advantage over existing methods.

**Questions:**

1. The method can highlight the anomalous regions. How is its segmentation performance on the detected anomaly areas and the classification accuracy on whether an image contains anomalies?

2. How about the storage cost (e.g., database size) and the inference speed of the proposed method?

---

### Official Review · Reviewer_V7K2 · 2025-11-01

**Soundness:** 3
**Presentation:** 3
**Contribution:** 2
**Rating:** 4
**Confidence:** 3

**Summary:**

This paper indicates that low-resolution views provide a better overall context, which helps in identifying when something is wrong. In contrast, high-resolution views offer more detailed boundary refinement, aiding in precisely locating where a defect exists. To take advantage of both perspectives, they proposed a multi-resolution fusion (MRF) strategy. This approach involves processing the image at multiple scales (resolutions), extracting features from each scale, and fusing them to combine the strengths of both low and high resolutions. They implemented this as a training-free method, named MRF-AD, meaning it does not require additional anomaly supervision. The method has shown competitive results on  MVTec AD 2.

**Strengths:**

1. It addresses the issue of scale sensitivity, that is, anomalies of varying sizes can be detected more effectively by considering multiple resolutions.
2. The fusion approach is straightforward conceptually and can easily be integrated with existing fundamental feature extractors.

**Weaknesses:**

1. Figure 1 does not effectively convey the motivation behind integrating recognition and refinement from different input resolutions.
2. Processing inputs at multiple resolutions and layers increases both computation and memory usage in direct proportion to the number of scales. The method may not be ideal for low-latency or resource-constrained settings.
3. The method depends on a high-quality pretrained encoder from a vision foundation model. If the features provided by the encoder are poor or not suitable for the anomaly domain, performance could decline clearly.
4. The claim of being “training-free” is somewhat overstated. Although the authors describe their approach as “training-free,” creating a memory bank from normal instances essentially involves modeling normality. There remains significant unsupervised processing and hyperparameter tuning involved.
5. Although fusion enhances results in their benchmarks, the paper focuses on specific scales and a particular backbone (DINOv2). The general applicability to all encoders and datasets is not fully demonstrated.

**Questions:**

1. Could the method be able to validated on non-industrial datasets, such as medical, satellite, or natural scene anomalies?
2. How sensitive is the method to the choice of resolution scales? Is there a systematic approach for selecting them for each dataset?
3. Can ablation study results directly demonstrate that low resolution contributes to recognition, while high resolution aids in boundary refinement?
4. Are there situations where multi-resolution fusion does not provide any advantages over single-resolution processing? If so, what do those situations look like?
5. How much does latency increase with each additional resolution scale? Is the method deployable on edge or embedded systems without GPU acceleration?

---

### Official Review · Reviewer_EGPJ · 2025-11-01

**Soundness:** 2
**Presentation:** 3
**Contribution:** 2
**Rating:** 4
**Confidence:** 4

**Summary:**

The paper introduces **MRF-AD**, a training-free method for unsupervised anomaly detection using features from a frozen DINOv2 encoder. The proposed method is motivated by that input resolution affects the model’s ability to detect and localize anomalies: 1) low-resolution inputs capture global context and improve recognition of anomalous regions; and 2) high-resolution inputs refine boundaries but may miss global coherence. Based on these two observations, MRF-AD builds multiple input-space views of each image at different resolutions and extracts features from several DINOv2 layers. For every (layer, resolution) pair, a memory bank of normal features is created. At test time, each feature is compared to its nearest neighbor in the corresponding bank, producing per-scale anomaly maps that are averaged into a final prediction. Experiments on MVTec AD v2 show competitive results, especially on the challenging $\mathrm{Test}_{priv,mix}$ split.

**Strengths:**

1. The “recognition vs. refinement” observation is intuitive and reasonable.
2. The proposed training-free approach is simple and shows good experimental results.
3. The ablation studies are comprehensive, evaluating layer fusion, resolution fusion, and background removal to verify each component’s contribution.
4. Implementation details appear to be sufficient for reproducibility.

**Weaknesses:**

1. The paper did not cite a highly relevant paper, **LogSAD** (CVPR 2025), which is also a training-free approach exploring multi-granularity fusion for anomaly detection.
2. The technical novelty is incremental. The proposed method integrates known elements, including training-free DINOv2 features, nearest-neighbor scoring, and score fusion, whereas it does not introduce a fundamentally new algorithmic idea. The contribution lies mainly in combining input-space multi-resolution fusion with multi-layer features. Relevant and similar ideas can be found in the LogSAD paper.
3. The presentation does not clearly describe whether the class label information is required at inference time. In particular, the description of memory-bank construction (Section 4.3) is ambiguous, and it is unclear if the memory banks are class-specific.
4. The presentation about the experimental results should clearly specify whether all baselines were evaluated under the same resizing protocol (i.e., 5 M pixels per image). Also, a runtime or memory-usage table is not provided, while dependence on BEN2 background removal slightly weakens the “training-free” narrative.
5. The description in the caption of Figure 1 does not match the four images.

**Questions:**

The authors are suggested to respond to those raised in **Weaknesses.**

**Additional questions**:
1. What are the total feature bank sizes and inference time for 1-, 3-, and 5-resolution settings?
2. How sensitive are results to the choice of resolution set (e.g., {0.3, 0.5, 0.7} vs. {0.4, 0.6})?

---

### Official Review · Reviewer_6zgn · 2025-11-01

**Soundness:** 3
**Presentation:** 3
**Contribution:** 3
**Rating:** 4
**Confidence:** 4

**Summary:**

The paper proposes Multi-Resolution Fusion (MRF) for training-free anomaly segmentation. The key idea is a “division of labor”: low-resolution inputs help with robust anomaly recognition by capturing global context, while high-resolution inputs sharpen boundaries with local detail. The method (MRF-AD) uses a frozen encoder, builds memory banks per layer and resolution, scores each view with nearest neighbors, then upsamples and averages the score maps to produce a scale-robust mask. On MVTec AD v2, it is competitive and achieves the best mean AU-PRO at threshold 0.05 on the challenging TEST-priv,mix split. The paper also offers clear ablations for layer fusion, resolution fusion, and background removal.

**Strengths:**

Clear, testable hypothesis about how low and high resolutions contribute differently, backed by intuitive visuals and ablation studies.
Solid empirical performance across splits, including the best mean AU-PRO (threshold 0.05) on TEST-priv,mix, with category-level wins and consistent improvements.
Training-free and easy to adopt in practice: no parameter learning beyond memory banks, compatible with standard approximate nearest-neighbor search and multi-layer fusion.
Clean presentation: the pipeline diagram aligns with the scoring procedure, labels and tables are easy to read, and the narrative is coherent for practitioners.

**Weaknesses:**

Multi-resolution/multi-layer fusion is standard; the advance is mainly a training-free packaging and careful eval, not a new objective/architecture.
No per-image latency/peak memory or accuracy–latency trade-offs; add a small profiling table and ANN sensitivity.
Relies on an external segmenter; show robustness with simple masks and at least one alternative segmenter (plus a few failure cases).
Evidence is concentrated on MVTec AD v2; even partial results on v1/VisA/BTAD/MPDD would help.
No defect-size stratification, and the drop with a larger backbone isn’t diagnosed; try basic normalization/cosine/PCA checks.

**Questions:**

1. Could you report per-image wall-clock latency and peak GPU memory for full fusion versus single-scale baselines, with a breakdown by ∣𝑆_res∣and ∣𝑆_layer∣, and include a small 𝑛_list / 𝑛_probe study to show accuracy–latency trade-offs?
2. Using your appendix distributions, can you report AU-PRO_0.05 for small/medium/large defects and indicate which resolutions contribute most in each regime?
3. If the external segmenter is replaced with a lightweight heuristic mask or an alternative pretrained segmenter, how does performance change, and can you provide a mosaic of failure cases?
4. For DINOv2-L, do cosine distance, feature whitening, or PCA compression prior to kNN mitigate the observed drop, and which factor appears causal?
5. How sensitive are results to 𝑆_res = {0.3, ... , 0.7}, and can a 2–3 scale subset recover most of the gains?
6. Please include results on MVTec v1/VisA?

---

### Note · Authors · 2025-11-13

I have read and agree with the venue's withdrawal policy on behalf of myself and my co-authors.